# The Influence of Socioeconomic Status (SES) and Processing Speed on the Psychological Adjustment and Wellbeing of Pediatric Brain Tumor Survivors

**DOI:** 10.3390/cancers14133075

**Published:** 2022-06-23

**Authors:** Maria Chiara Oprandi, Viola Oldrati, Claudia Cavatorta, Lorenza Gandola, Maura Massimino, Alessandra Bardoni, Geraldina Poggi

**Affiliations:** 1Neuro-Oncological and Neuropsychological Rehabilitation Unit, Scientific Institute, IRCCS E. Medea, 23842 Bosisio Parini, Italy; viola.oldrati@lanostrafamiglia.it (V.O.); alessandra.bardoni@lanostrafamiglia.it (A.B.); geraldina.poggi@lanostrafamiglia.it (G.P.); 2Medical Physics Unit, Fondazione IRCCS Istituto Nazionale dei Tumori, 20133 Milan, Italy; claudia.cavatorta@istitutotumori.mi.it; 3Pediatric Oncology Unit, Department of Medical Oncology and Hematology, Fondazione IRCCS Istituto Nazionale dei Tumori, 20133 Milan, Italy; lorenza.gandola@istitutotumori.mi.it (L.G.); maura.massimino@istitutotumori.mi.it (M.M.)

**Keywords:** childhood cancer, adolescents, young adults, survivors, psychosocial, mental health, vulnerable groups, processing speed, socio-economic status, mediation analysis

## Abstract

**Simple Summary:**

Processing speed (PS) is one of the most impaired functions in pediatric brain tumor survivors (PBTSs) and it has been linked to difficulties in their psychological functioning, together with other non-insult-related risk factors, such as socio-economic status (SES). Given the psychological adjustment difficulties observed in PBTS, the aim of the current study was to explore the relationship between SES and psychological functioning, measured with the Child Behavioral Checklist (CBCL) and the Strengths and Difficulties Questionnaire, and considering the contribution of PS as a mediator. The results demonstrated that the influence of SES on the CBCL total index was mediated by PS. Furthermore, PS was found to have a mediating effect on the SES–internalizing problems relationship but not on the SES–externalizing problems relationship. These findings suggest that PS may be a rehabilitation target to prevent psychological distress and should be addressed, especially for PBTSs who live in a disadvantaged situation.

**Abstract:**

(1) Background: The relationship between processing speed (PS) and psychological adjustment in the healthy population is well established, as is that between low socio-economic status (SES) and psychological distress. While PS is one of the most impaired functions in pediatric brain tumor survivors (PBTSs), previous research has demonstrated that low SES may be a predictor of increased psychosocial risk in PBTSs. Given the psychological adjustment difficulties observed in PBTS, in the current study we aimed to explore the relationship between SES and psychological functioning, considering the contribution of PS as a mediator. (2) Methods: demographic and clinical data of 80 children (age range: 4–17 y.o.) were retrospectively collected. Psychological measures were the parent-compiled versions of the Child Behavioral Checklist (CBCL) and the Strengths and Difficulties Questionnaire (SDQ). Mediation analysis models were performed on psychological measures with and without the inclusion of covariates. (3) Results: The influence of SES on the CBCL total index was mediated by PS. Furthermore, PS was found to have a mediating effect on the relationship between SES and internalizing problems but not on the relationship between SES and externalizing problems. (4) Conclusions: The results suggest that PS may be a rehabilitation target for the prevention of psychological distress and should be addressed especially for PBTSs who live in a disadvantaged situation.

## 1. Introduction

According to the World Health Organization [1], psychosocial adjustment, together with physical condition, mental status and wellbeing, plays an important part in the definition of health, referring to “people’s capacity to adapt to the environment, which implies that the individual has sufficient mechanisms to feel good, integrate, respond adequately to the demands of the environment, and achieve his or her objectives” [2].

Several pieces of evidence have demonstrated that pediatric brain tumor survivors (PBTSs) exhibit more difficulties in psychological adjustment when compared to their healthy peers (i.e., their siblings), but also in respect to other oncological patients, such as leukemia survivors [3,4,5,6,7,8].

Among the main complications, PBTSs experience internalizing problems, related to withdrawal, anxiety, depression [9,10], somatization [4], difficulties in social functioning and acceptance, especially by peers [11,12,13]. Moreover, pediatric cancer survivors may also develop in a low percentage of cases post-traumatic stress disorder symptoms (4.7%–21%) [14,15] and suicidal ideation (6–8%) [16]. Parent- and teacher-compiled reports also suggest the presence of externalizing problems in some cases, such as antisocial behaviors, aggression, and low behavioral regulation [11]. Moreover, motor impairment may cause difficulties in the ability to perform physical roles, and survivors have poorer expectations of future life satisfaction [4].

These difficulties typically arise in the first 12 months after the tumor diagnosis and may continue even years after the end of treatment [11,17,18,19]. For some patients, complications last until adulthood, leading to reduced quality of life and difficulties in independent living [18,20,21].

Several factors have been shown to influence psychosocial adjustment, including processing speed (PS) and socio-economic status (SES), both in the healthy population [22,23] and in pediatric cancer survivors [24,25]. Moreover, a link between low SES and poor cognitive functioning (i.e., PS) has been found in pediatric oncological patients [26].

PS is the speed at which mental operations are performed [27]. Slow PS has been found to be associated with different neuropsychiatric disorders, such as attention deficit-hyperactivity disorder (ADHD) [28], especially the inattentive type [29] and autism spectrum disorder (ASD) [29,30]. PS impairment has also been observed in patients at risk of developing psychotic disease [29,31] and in those with socio-emotional difficulties (e.g., depression) [32]. Accordingly, it has been suggested that, in PBTS, poor cognitive and neuropsychological functioning, including but not limited to impaired PS, may contribute to a high risk of psychological adjustment problems [33,34].

SES is a multi-dimensional construct, including economic resources, education, and the rank of a subject on the socioeconomic scale [35], and it has been related to several health outcomes [36,37,38,39]. The link between SES and psychological distress is well established in the healthy population: higher levels of psychopathological problems, both internalizing and externalizing [40,41], are often reported in children coming from families with a low SES [42,43], with a percentage three times higher than their peers from high-SES families [42].

In the pediatric oncological population, SES and low parent education have been related to high mortality rates and decreased psychological adjustment [25,44,45,46,47].

The mechanisms by which SES exerts its influential role on cognitive and psychological functioning are still a matter of debate. Social causation accounts propose that SES could determine brain differences through environmental influences shaping brain structure, function, and development. Potential negative pathways for this mechanism include, among others, exposure to toxic agents, cognitive under-stimulation, unhealthy nutrition, inadequate parenting styles and transient or chronic effects [48,49,50,51,52,53,54,55], such as chronic stress. According to this view, highly or lowly SES-linked conditions could determine neural correlates [56], affecting some neural systems more than others [55]. Interestingly, language and executive functions seem to be more greatly impacted by lower SES, although they are not the only domains affected (see Farah, 2017 for a review [56]). On the other hand, social selection accounts propose that brain differences under genetic control should be recognized as the cause of SES differences. From this standpoint, the affected functions cause lower or non-adequate performances, with consequences on SES [55], in terms of, for example, academic and work opportunities, social support and stimulating and enriching environments, creating a declining vicious circle. Distinct possible relations among the causes and consequences of SES and its neural and functional correlates have been proposed: moderation of the link between the brain and behavior by SES, the mediation of life outcomes and behavioral consequences of SES by brain structures and functions, and the mediation of SES–brain relations by proximal factors (such as the level of cognitive stimulation, nutrition, parenting styles, chronic stress, etc.) associated with SES (see Farah, 2017 for a review [56]). In the context of this study, we examined these relations adopting the view of brain structures and functions as mediators of the effects of SES on life outcomes.

Unfortunately, there is little agreement on how SES should be operationalized in pediatric health research [49,57,58,59]. However, several studies have suggested that collecting composite measures, incorporating several domains of information into a singular quantity, is more useful than relying on proxy indices depending on a single domain [60]. Moreover, many studies have not reported their rationale for the choice of using a specific SES measure [61], which may complicate the interpretation of findings.

The Hollingshead Four Factor Index (HI) is one of the most frequently used measures of SES [62] and it appears in the literature on the relationship between socio-economic factors and symptoms in different clinical populations, such as dyslexia in adult patients [63], preterm newborns [64] and psychiatric disorders, including major depression [65,66]. Despite this, some researchers have raised concern about specific aspects of this index, such as the calculation of the index in cases of divorced or widowed spouses [62]. Indeed, one major issue raised about the use of HI is that this index is often computed based on the education and occupation of the primary earner of the family, who is usually the father, so that maternal education and occupations are frequently not included in the overall index whenever the mother is a housewife or is unemployed [67].

In contrast, existing evidence seems to suggest that maternal education may play a significant role in child psychological adjustment. Indeed, maternal education was found to be the best predictor of maternal warmth and responsivity in a clinical sample of children with oppositional defiant disorder [67]. Positive associations were found between maternal education and the quality of parenting behavior [68,69,70,71], emotional availability [72] and sensitivity [73], meaning the caregiver’s capacity to evaluate the type of care required by their child [74], which are all competences that are able to modulate psychological adjustment. On a related note, lower maternal education has been found to be associated with a shorter survival rate in children diagnosed with brain tumors [75].

Moreover, many mothers drop out of their jobs once they have children [67], or return to work later than fathers, especially when their child has a severe physical health condition [76] such as a brain tumor. Additionally, mothers are often the primary caregiver, who spends more time with the child [67].

Due to the reasons discussed above, information pertaining to the maternal socio-economic status should be considered in research related to child health-related outcomes. Finally, recommendations about improved SES computation methods have been made in the literature, suggesting that this index should be based on a group-specific approach and that it should include “(1) plausible explanatory pathways and mechanisms, (2) as much relevant socioeconomic information as possible, (3) specifying the particular socioeconomic factors measured (rather than SES overall), and (4) systematically considering how potentially important unmeasured socioeconomic factors may affect conclusions” [36].

According to Braveman’s schema (1), we based the current study on previously examined pathways between SES and psychological adjustment and between SES and PS; (2) we considered information about the occupational and educational status of the mother, even when she was a housewife, as well as the father of the patients included in our sample; (3) we explored the importance of the collected factors included in the overall measure; and finally (4) we also considered other clinical disease-related factors that may have an important effect on the current results (i.e., age at diagnosis and time since diagnosis).

In accordance with the data emerging from the relevant literature on both healthy subjects and PBTSs, the main aim of the present study was to investigate the relationship between SES and psychological functioning, considering the contribution of PS as a mediator. Understanding the extent of the relationship between SES and psychological adjustment and its potential mediation by PS could help clinicians to identify the most at-risk subjects and to establish better interventions for this clinical population.

## 2. Materials and Methods

### 2.1. Participants

The overall sample was composed of 80 patients between 4 and 17 years of age at assessment with a diagnosis of brain tumor, admitted to a pediatric rehabilitation center in Italy (La Nostra Famiglia, Scientific Institute I.R.C.C.S. E. Medea, Bosisio Parini, Italy). The clinical characteristics of the patients are listed in Table 1.

The sample was composed of patients coming from two groups, which have already appeared in two previous papers. Among these, 50 patients out of 78 were selected from the sample examined in Oprandi et al., 2021 [77], and 30 patients out of 45 were selected from the sample examined in Cavatorta et al., 2021 [78]. Forty-three patients were excluded as they fell outside the age range, defined according to the psychological measures selected to answer our research question. Hence, patients below 4 years of age and assessed by means of the Griffiths Mental Development Scales were not included, as this test yields IQ scores that are not directly comparable with the Wechsler scales. Moreover, as the CBCL and the SDQ parent reports can be administered until 18 and 17 years of age, respectively, we could not include patients older than 17 years of age. Furthermore, including patients ≥ 18 years could have complicated the interpretation of the results, as typically older patients have different problems and needs.

Inclusion criteria for the selection of the patients were: (i) diagnosis of a brain tumor; (ii) age between 4 and 17 years at assessment; (iii) absence of severe sensorial, motor, and articulatory deficits that could impede the cognitive assessment; and (iv) the presence of all the selected cognitive and psychological measures. Patients were excluded if they had a pre-existing neurodevelopmental disorder or disability, a diagnosis of a pervasive developmental disorder, a diagnosis of neurofibromatosis or if they developed posterior fossa syndrome after neurosurgery.

Approval was received from the local ethical standards committee on human experimentation at the Scientific Institute I.R.C.C.S. E. Medea. The sample from Oprandi et al., 2021, was part of an observational study. Hence, the Ethics Committee of Scientific Institute I.R.C.C.S.E. Medea only required notification about the study (identification number: 03.2021 Oss; date of approval: 14 April 2021). The sample from Cavatorta et al., 2021, was collected for a retrospective and a prospective study (Research no. 14486, date of approval: 27 February 2017), and written informed consent was required from the parents/caregivers prior to study enrollment. The study was conducted in agreement with the principles expressed in the 1964 Declaration of Helsinki.

### 2.2. Measures

#### 2.2.1. Demographic and Clinical Information

Information on the medical variables was collected from the patients’ charts. Age at diagnosis, age at assessment, age at radiotherapy (for those who underwent such a treatment) and time since diagnosis (the time between diagnosis and assessment) were collected or calculated in months. The histopathological type of tumor was classified as astrocytoma, medulloblastoma, ependymoma or other diagnosis (such as primitive neuro-ectodermal tumor (PNET), craniopharyngioma, glioblastoma, atypical teratoid rhabdoid tumor, choroid plexus tumor, pineoblastoma or brainstem glioma). The tumor location was rated as supra- or infratentorial. Moreover, the number of neurosurgeries was collected, together with chemo- and radiotherapy (craniospinal vs. focal radiotherapy). The grade of tumor was divided into mild (grades I and II of the 2016 World Health Organization classification of tumors of the central nervous system [79]) or severe (grades III and IV). A history of hydrocephalus was recorded as present or not present.

#### 2.2.2. Cognitive Functioning

From the Wechsler age-appropriate scale (Wechsler Preschool and Primary Scale of Intelligence-3rd Edition, Wechsler Intelligence Scale for Children, 3rd and 4th Edition) the following indexes were retrieved:The full scale intelligence quotient, which is a measure of global cognitive functioning, and;The processing speed index (PSI), which is a measure of the ability to respond promptly and focus attention on a task.

Scores are reported as age-corrected standard scores with a mean of 100 and a standard deviation of 15. Lower scores represent worse performance [80,81,82]. In accordance with the routine clinical practice, the neuropsychological evaluation and test scoring were carried out by trained psychologists.

#### 2.2.3. Psychological Adjustment and Behavioral Functioning

The Child Behavioral Checklist (CBCL) is a parent-reported questionnaire composed of 113 items to evaluate child behavioral, social and emotional competence in the past 6 months. Each item is scored on a three-point scale: 0 = not true, 1 = somewhat or sometimes true, 2 = very true or often true. The questionnaire yields eight subscale scores (‘anxious/depressed’, ‘withdrawn/depressed’, ‘somatic complaints’, ‘social problems’, ‘thought problems’, ‘attention problems’, ‘rule-breaking behavior’ and ‘aggressive behavior’) and three global scores: internalizing (e.g., fearful, shy, anxious, inhibited), externalizing (e.g., aggressive, antisocial, under-controlled) and total behavioral problems. The anxious/depressed and withdrawn/depressed sub-scales form the internalizing scale, whereas the aggressive behavior and rule-breaking behavior sub-scales form the externalizing scale. Two forms are available, based on the child’s age: 1.5–5 years and 6–18 years.

Raw scores were converted into T-scores: for the eight subscales, scores below 65 fell into the normal range; scores between 65 and 69 were in the borderline range, whereas scores above 70 were considered in the clinical range. For the three main scales, scores above 63 were considered in the clinical range, whereas the borderline range was between 60 and 63 [83]. Only the parent-compiled report was used in the present study.

The CBCL has been shown to have strong reliability and validity in clinical and normal populations. Internal consistency and reliability is between 0.57 and 0.71 for internalizing problems, 0.70 and 0.86 for externalizing problems, and 0.69 and 0.82 for total problems [83].

The scale measures for this instrument were standardized by age and sex using the available global cross-cultural norms.

The CBCL is a widely used, empirically derived and supported tool in pediatric cancer patients and PBTSs [84]. According to the parent-reported form results, internalizing behavior problems were more common in PBTS than externalizing problems, which were less frequent [85,86,87], but these results are not always in accordance with one another.

The relevant literature suggests that some of the results derived from specific subscales should be taken with caution. Indeed, regarding the somatic complaint subscale, headaches, vomiting or nausea might be also tumor-related symptoms and not an expression of psychological distress. Furthermore, high scores in the social problems subscale might not be related to social difficulties, but rather to the lack of access to social interactions, because of long hospitalization or frequent absences from school or the presence of motor/sensory sequelae [84].

#### 2.2.4. Psychological Wellbeing

The Strengths and Difficulties Questionnaire (SDQ) is a 25-item-questionnaire for children, available in three formats: a parent-compiled version for children of 3–4 years old and another for children of 4–17 years old, and a self-reported version for children of 11–17 years old. Only the parent-compiled report (4–17 y.o.) was used in the present study. The questionnaire yields five subscale scores of five items each: emotional symptoms, conduct problems, the hyperactivity/inattention subscale, peer relationship problems and prosocial behavior. Moreover, an overall score is available; the total difficulties score implies whether and to what extent the child’s problems influence his/her daily behavior. Higher scores indicate more problems, except for the prosocial behavior subscale, in which higher scores indicate greater social competence.

The SDQ has good psychometric properties. Its reliability is generally satisfactory: its internal consistency is 0.73, its cross-informant correlation is equal to 0.34, and its retest stability after 4 to 6 months is 0.62. The validity, derived from the association between high SDQ scores and disorders diagnosed with the Diagnostic and Statistical Manual-IV, highlighted the increase in psychiatric risk, with odds ratios of approximately 15 for parent and teacher SDQ scales and of approximately 6 for self-reported SDQ scales [88].

The SDQ has been used in research in PBTS. In a study by Upton and colleagues, parents rated their children as having more behavioral and emotional problems than would be expected from population norms, in all the SDQ subscales, except the conduct problem scale [89]. Moreover, Williams and colleagues highlighted the potential usefulness of the SDQ as a screening tool to recognize childhood cancer survivor psychosocial difficulties [90].

#### 2.2.5. Socio-Economic Status Index

Socio-economic status was calculated according to the Four Factor Index indications, collecting information on the education and occupations of both parents [91]. Occupations were rated on a 9-point scale, in which 9 stands for high-income occupations (such as higher executives, proprietors of large businesses and major professionals) and 1 for low-income occupations (i.e., farm laborers or menial service workers). Education was rated based on the years of schooling, on a 7-point scale, in which 7 indicated graduate professional training and 1 indicated less than a seventh-grade education (in Italy this is equivalent to the first year of middle school, corresponding to 11 year old students).

The original version of the index was obtained by weighting and summing the education and occupation scores as follows: the occupation score multiplied by five was added to the education score multiplied by three. When both parents were gainfully employed, the overall index was calculated as the average of the two earners. Otherwise, in cases when the mother was a housewife, only the education and occupation of the father was used to obtain the overall index. Indications for the computation of the index in specific cases (i.e., separated or divorced persons, widows, or widowers) were given by Hollingshead [67,91]. The SES range was between 8 and 66.

However, considering the issues raised on the use of HI in its original version, in the context of the present study, the decision regarding how to calculate the SES index was guided by the exploration of correlations between the different information gathered on the educational and occupational levels of the mothers and the fathers of patients and the CBCL and SDQ main scales. The following indexes were considered: the mother’s occupational level; the father’s occupational level; the family’s occupational status; the mother’s educational level; the father’s educational level; the family’s educational status; the CBCL-guided mother’s occupational index and the CBCL-guided father’s occupational index. Table 2 reports the correlation results.

Overall, negative correlations indicating increasing symptoms with decreasing parental occupation/education level were found. In more detail, as only indexes on the occupational level relating to the mother, but not to the father, significantly correlated (fully or marginally) with three out of four psychological outcomes—i.e., CBCL total score, CBCL externalizing score and SDQ total score—we decided to change the standard calculation procedure of SES as described in Hollingshead [91], by including the educational level of the mother, even if she is a housewife. As already mentioned, according to the standard procedure, if the mother is a housewife, then she has to be considered unemployed and thus removed completely from the calculation of the total index, which appraises exclusively the educational and occupational level of the father. However, the limits of this index have been highlighted (see Introduction). In an attempt to overcome these limits, we decided to create an SES index which includes the educational level of the mother, even if she is a housewife, to at least partially decrease the underestimation of the impact that the mother’s education and occupation has been shown to exert on a child’s psychological adjustment [67,68,69,70,71,72,73]. In that case, the overall index is calculated according to the following formula:SES = (Mother’s Education * 3) + [(Father’s Education * 3) + (Father’s Occupation * 5)]/2

#### 2.2.6. Data Diagnostics and Statistical Analysis

In order to verify the relational assumptions to run mediation analysis, Pearson correlations were analyzed for the continuous outcome variables (i.e., SES, PSI and questionnaire scores). Hence, we verified that the causal variable (SES) was correlated with the mediator (PSI) (test path a) and that the mediator affected the outcome variables (questionnaire scores) (test path b). The selected SES index was correlated to PSI (r = 0.25, *p* = 0.03), as indicated by the higher SES values in patients with higher PSI scores. PSI was significantly correlated with CBCL total score (r = −0.37, *p* = 0.001), the CBCL internalizing score (r = −0.34, *p* = 0.002) and the SDQ total score (r = −0.44, *p* = 0.0001), whereas the relation with the CBCL externalizing score was nearly significant (r = −0.21, *p* = 0.06). For all scales, lower PSI was associated with more pronounced symptoms. The selected SES index was significantly correlated to the CBCL total score (r = −0.26, *p* = 0.02) and the CBCL externalizing scale (r = −0.28, *p* = 0.01), whereas it was only marginally correlated to the CBCL internalizing scale (r = −0.19, *p* = 0.09) and the SDQ total score (r = −0.19, *p* = 0.08). As indicated by the negative *r* coefficient, as the SES index tends to increase, symptoms tend to decrease. Table 3 reports the correlation parameters between SES, PSI and all CBCL and SDQ scales and subscales.

Moreover, in light of previous research showing a relationship between both the age of the child at diagnosis and the time from treatment to evaluation and psychological adjustment, such a relationship was tested in our sample. Correlation analysis indicated that although the time from treatment to evaluation did not correlate to any questionnaires’ scales, the age of the child at diagnosis did correlate to the SDQ total score (r = 0.22, *p* = 0.05) and marginally to the CBCL internalizing score (r = 0.22, *p* = 0.06), with increasing symptoms at an older age at diagnosis. Hence, mediation models on the questionnaires’ main scales were also computed, including the age at diagnosis as covariate to further understand the potentially predictive value of this variable. Furthermore, sex and histopathological tumor type were also added into the mediation models as covariates, respectively, as demographic and clinical factors of interest.

Separate mediation models were conducted with SES predicting CBCL and SDQ scale and subscale scores via PSI. According to the empirical estimates of sample size for the bootstrapping procedure computed by Fritz and MacKinnon (2007), the sample size necessary to achieve a power of 0.8 is 78, when considering a medium effect size for the a and b paths. These estimates were based on previous meta-analytic accounts [92,93] finding small-to-medium effects of SES on executive functions in children and evidence of a small-to-medium relation between processing speed and psychological problems [22], varying according to the specific measure considered. Thus, a sample of 80 patients was considered sufficient to test the mediation effects between the variables of interest.

In accordance with previous accounts [94,95], an examination of indirect effects was carried out as well in the case of a non- or marginally significant total effect of the independent variable on the identified dependent variables. Mediation analyses, with and without covariate adjustments, were conducted using the PROCESS macro for R [96]. PROCESS calculates indirect effects by adopting a bootstrapping resampling procedure. Data were sampled 10,000 times with replacements to calculate a 95% bootstrap confidence interval. Completely standardized effects (β) were computed for the total, direct and indirect effect models and used as an indication of effect size. R software (version 4.0.3; R Foundation for Statistical Computing, Vienna, Austria) was used to perform all of the statistical analyses in this study.

## 3. Results

As regards the CBCL, the descriptive statistics showed the following percentages of patients falling within the clinical range (Figure 1a): CBCL total: 17.5%; internalizing score: 23.8%; externalizing score: 6.3%; anxious/depressed: 8.8%; withdrawn/depressed: 11.3%; somatic complaints: 8.8%; social problems: 7.5%; thought problems: 7.5%; attention problems: 5.0%; rule-breaking behavior: 2.5%; aggressive behavior: 1.3%. As for the SDQ, the following percentages of patients were observed to fall within the clinical range (Figure 1b): SDQ total: 10.0%; emotional problems: 15.0%; conduct problems: 10.0%; hyperactivity: 27.5%; peer problems: 25.0%; prosocial behavior: 3.8%. On the basis of the indications provided by the tests considered in the current study and by psychological counselling sessions, 25 patients (31.25%) out of 80 received the indication to undertake psychological support or to continue the one already started. The remaining were suggested to closely monitor their psychological functioning in the yearly follow-up.

Below, we report the results of the mediation analyses. As regards the total CBCL score, the mediation analysis showed a significant total effect of SES on the CBCL scale (β = −0.26, SE = 0.12, *p* = 0.02). However, the direct effect expressing the impact of SES on the dependent variable after “controlling for” PSI was no longer fully significant (β = −0.19, SE = 0.12, *p* = 0.08). The completely standardized indirect effect of PSI on the CBCL scale was found to be significant (β = −0.08; SE = 0.05; CI: −0.19, −0.003), indicating that PSI mediated the effect exerted by the SES index on the CBCL total score (Figure 2a). In the model with covariate adjustment, neither age at diagnosis (*p* = 0.56), sex (*p* = 0.57) nor histopathological tumor type (*p* = 0.71) were found to be significant predictors. The model on the CBCL internalizing score showed a marginally significant total effect of SES on the CBCL scale (β = −0.19, SE = 0.13, *p* = 0.09). However, the direct effect expressing the impact of SES on the dependent variable after “controlling for” PSI was far from significant (β = −0.11, SE = 0.12, *p* = 0.31). The completely standardized indirect effect of PSI on the CBCL scale was significant (β = −0.07; SE = 0.05; CI: −0.19, −0.003), indicating that PSI mediated the (weak) effect exerted by the SES index on the CBCL internalizing score (Figure 2b). In the model with covariate adjustment, neither age at diagnosis (*p* = 0.09), sex (*p* = 0.48) nor histopathological tumor type (*p* = 0.75) were found to be significant predictors.

The model on the CBCL externalizing score showed a significant total effect of SES on the CBCL scale (β = −0.28, SE = 0.09, *p* = 0.01). The direct effect, indicating the impact of SES on the dependent variable after “controlling for” PSI, was still significant (β = −0.24, SE = 0.09, *p* = 0.04). The completely standardized indirect effect of PSI on the CBCL scale was not significant (β = −0.04; SE = 0.05; CI: −0.12, 0.02), indicating that PSI did not mediate the effect exerted by the SES index on the CBCL externalizing score (Figure 2c). In the model with covariate adjustment, neither age at diagnosis (*p* = 0.83), sex (*p* = 0.16) nor histopathological tumor type (*p* = 0.65) were found to be significant predictors.

Finally, as regards the SDQ total score, the mediation analysis showed a marginally significant total effect of SES on the questionnaire scale (β = −0.19, SE = 0.06, *p* = 0.08). However, the direct effect expressing the impact of SES on the dependent variable after “controlling for” PSI was far from significant (β = −0.09, SE = 0.06, *p* = 0.38). The completely standardized indirect effect of PSI on the SDQ scale was significant (β = −0.10; SE = 0.06; CI: −0.23, −0.01), indicating that PSI mediated the (weak) effect exerted by the SES index on the SDQ total score (Figure 2d). In the model with covariate adjustment, age at diagnosis was found to be a significant predictor (*p* = 0.04), whereas sex (*p* = 0.32) and histopathological tumor type (*p* = 0.41) were not. A summary of the mediation model results for the CBCL and SDQ main scales (with covariate adjustment) and subscales (without covariate adjustment) is presented in Table 4.

## 4. Discussion

In the current study we examined the relationship between SES and psychological functioning, considering the contribution of PS as mediator, in a group of PBTSs.

First, the selection of the SES index was based on the exploration between the different information gathered on the educational and occupational level of the mothers and the fathers of patients and CBCL and SDQ main scales. As already mentioned in the Methods section, according to the standard procedure, if the mother is a housewife, then she has to be considered unemployed and is thus removed completely from the calculation of the total index, which appraises exclusively the educational and occupational level of the father. However, the limits of this index have been highlighted (see Introduction). In an attempt to overcome these limits, we decided to create an SES index which included the educational level of the mother, even if she was a housewife, to at least partially decrease the underestimation of the impact that the mother’s education and occupation has been shown to exert on a child’s psychological adjustment [67,68,69,70,71,72,73].

Preliminary analysis of the data showed that the SES score and PS were both associated with the severity of psychological and behavioral adjustment difficulties displayed by PBTSs, measured with both the CBCL and SDQ. More precisely, PS was found to be negatively correlated with internalizing problem scores, including anxiety and depressive symptoms, withdrawal and somatic complaints, as well as other difficulties within the socio-emotional domain and thought problems.

These findings of a lower PS in patients with more pronounced internalizing problems could be explained by direct or indirect consequences of the tumor and tumor-related treatments. Indeed, from a mechanistic point of view, PS is mainly linked to white matter integrity, which is extremely important for the diffusion of action potentials but particularly vulnerable to toxic and physical agents, such as radiation therapy [97,98]. Research on the associations between white matter and psychological outcomes in adolescents with major depression indicated a pattern of reduced white matter functioning in structures or tracts located in or passing by the frontal and prefrontal cortices [99], which are the last to be fully myelinated and which thus might be more vulnerable to toxic agents such as the tumor or the oncological treatments [97,98].

As regards the relation between PS and social competences which emerged from the correlation analysis, the former could influence the latter in different ways. First, both the cognitive components of social skills [100] and social adjustment competences are supported by the PS [101,102] and therefore could be damaged in PBTSs. Second, PBTSs with slower PS may tend to be more isolated, self-excluding themselves from social situations.

The link between PS and thought problems that emerged here is in line with a previous account of PS impairment, but not executive function deficits, in children with obsessive compulsive disorder [103].

On the other hand, SES was found to be negatively correlated with externalizing problems, referring to rule-breaking and aggressive behavior, as well as with attention problems and hyperactivity. The current results are along the same lines as previous findings reporting that a non-insult-related risk factor, such as SES, is associated with more social and behavioral problems in PBTS [104,105].

In regard to the main aim of the current study, the mediation analyses showed that PS “fully” mediated the relationship between SES and the CBCL total score, but did not mediate its relationship with the CBCL externalizing index. As regards both the CBCL internalizing index and the SDQ total score, the analysis showed that the weak link between SES and these two indexes was further weakened after controlling for the mediating role of PS.

The finding of a mediation by PS of the link between SES and the general psychological adjustment in PBTSs is in line with similar findings emerging in the healthy pediatric population [106]. Alongside the well-established relationship between low SES and increased risk of psychopathology [23], a study by McNeilly and colleagues [106] found that children and adolescents from lower-SES households were more likely to exhibit greater executive function difficulties and that such difficulties partially mediated the effect of SES on both externalizing and internalizing psychopathological symptoms. Importantly, they further demonstrated that this mediation was not explained by a co-occurring form of adversity such as the exposure to violence. In a similar vein, the current findings indicated that PS mediated the relationship between SES and the CBCL total score, which includes symptoms referring to both the internalizing and the externalizing spectra. Indeed, PS is a lower-level skill which is hypothesized to be subservient to higher-level skills, such as executive functions [107]. Injuries occurring at any time during brain development may lead to severe impairments in executive functions, which are significantly related to a greater risk of emotional and behavioral difficulties [108] as well as deficits in social information processing and social competence [100,109,110,111].

Notably, the result of the SDQ total score, which primarily focuses on problems that may be considered conceptually closer to the externalizing symptomatology, were along the same lines, although this measure was only weakly related to the SES, as shown by the total effect model.

However, when considering internalizing and externalizing scores separately, the analyses showed that PS mediated the weak relationship between SES and the former, whereas it did not mediate the stronger relationship which emerged between SES and the latter. Such differential effects may, first of all, depend on the specific challenges faced by PBTSs, as discussed below, but may also be due to the choice of the mediator. In fact, although poor executive functions have been more consistently associated with externalizing problems than with internalizing disorders [112,113], a low PS has been observed in pediatric patients displaying emotional problems, including major depression disorder [32]. In keeping with this, in the present study withdrawn-depressed was the only subscale used for the computation of the internalizing index score, and of which the total effect showed a near-to-significance effect of SES, found to be “fully” mediated by PS. Moreover, it has been suggested that, given the prevalence of PS deficits across a range of disorders, processing speed should be evaluated as a specific construct in transdiagnostic research frameworks [114]. Furthermore, the use of a performance-based measure as a mediator (such as PS), as compared to a parent-compiled report (as in McNeilly [106]), may have grasped a different component of a broader concept of cognitive capacity.

Notably, several studies have reported that internalizing problems are more frequent in PBTSs than externalizing behaviors [10,87,99]. Accordingly, we observed that the percentage of patients with clinically relevant internalizing symptoms was higher than the percentage of those displaying externalizing problems (see Results; Figure 1a). We speculate that both the fact that PBTSs appear to be more susceptible to developing internalizing disorders and the fact that impaired PS could be considered a component of overall psychopathological severity [22] may explain the mediating role of PS which has emerged here.

The absence of a mediating effect of PS on the externalizing scale may be due to the equally weak link found in our sample between PS and behavioral problems (especially the CBCL aggressive and SDQ conduct problem subscales). Indeed, our patients reported that they suffered more from internalizing than externalizing problems, as highlighted above. Moreover, other important factors eluding the cognitive functioning of the individual have been recognized to pose a greater risk for the development of externalizing problems. Several studies have suggested that both paternal and maternal behavior may influence, or otherwise prevent, the offspring from developing externalizing problems [115]. For example, low levels of parental responsiveness may suggest insensitive and rejecting behavior of parents towards their children [116]. Furthermore, parents displaying an inadequate kind of control, such as harsh (meaning physical or verbal punishment) [117] or psychological authority (such as manipulation, guilt induction, shaming and conditional love) [118] can lead to negative psychological outcomes for the child.

If we consider each subscale separately, the analyses showed that CBCL attention problems and the SDQ hyperactivity were the only subscales mediated by PS. It may be relevant to point out that the former was excluded from the computation of both the internalizing and the externalizing indexes, and that hyperactivity is used together with all the other SDQ subscales to constitute the questionnaire’s overall score. Our result may be linked to previous reports of an association between PS impairment and inattentive symptomatology in pediatric patients [28,119].

Overall, the present results suggest that PS mediates the effect of low SES on psychological difficulties in PBTS, echoing evidence on the mediating role exerted by individual cognitive reserves on psychopathology in both healthy young individuals [106] and children facing early-childhood deprivation [120]. In more detail, PS was found to “fully” mediate the effect of a general proxy of psychological difficulties, including both internalizing and externalizing symptoms, as well as attention problems and hyperactivity. On the other hand, PS was not found to be able to mediate the impact of SES on the externalizing symptoms. These results were confirmed in additional analyses carried out to investigate the potential impact of age at diagnosis, inserted in the models as a covariate. Importantly, age at diagnosis was found to be a marginally significant variable contributing to the internalizing problems. One possible explanation for this outcome lies in the fact that such difficulties may be more strongly linked to the awareness of the disease and its consequences: the older the patients at the time of the diagnosis, the more awareness he/she has about his/her health condition [121], possibly leading to the development of emotional problems.

Nevertheless, it has to be noted that small-to-medium β coefficients emerged in the total effect models (path c) of the CBCL total and CBCL externalizing scale, and small β coefficients emerged in the models of the CBCL internalizing and the SDQ total scale. In a similar vein, β coefficients ranging from small to medium were found for the relation between PSI and the questionnaire scales (path b). These estimates were lower than those expected on the basis of the literature review and which were used in the power analysis calculation. Hence, it cannot be excluded that non-significant findings on the mediation of SES on psychological adjustment via PSI may depend on the fact that a sample of 80 participants was not sufficiently large to detect a mediation effect.

### 4.1. Clinical Implications

Taken together, the present findings suggest that the SES of PBTSs’ families should be taken into account when planning interventions aimed at preventing the psychological and psychopathological sequelae of the disease. More specifically, interventions in favor of low-SES families should be implemented to improve their conditions, in accordance with socio-health services, focused on finding employment for unemployed parents or on giving support to the mother in the household, in order to alleviate the burden of care of the disease [122].

Further, clinical attention should be devoted to PS as a potential target of intervention in light of its mediating effect on the relationship between SES and psychopathology. Existing training, particularly computerized interventions targeting PS, working memory and attention, have shown promising results [123,124].

Nevertheless, major concerns have arisen regarding the ability of these programs to generalize the improvement to non-trained and daily life competences, such as academic performance [94].

Finally, different interventions in children from poor homes were demonstrated to improve performances in tasks involved in different functions supported by PS, such as executive functions [125,126,127,128,129,130] selective attention [126,128], inhibitory control, working memory, flexibility and planning [130]. Different modalities of training were tested: multi-modular intervention programs [125,126,131], training activities integrated into school curricula [132,133] and parenting interventions [134,135].

Interestingly, the evidence from these studies suggested that interventions dedicated to multiple targets, i.e., both children and their parents, are highly effective in improving the cognitive functioning of the former and the caregiving behavior of the latter [126]. However, the findings also demonstrated that not all children who participate in the training interventions improved their performance uniformly, potentially because of individual characteristics and/or contextual factor variability [128,136], which need to be further investigated.

### 4.2. Limitations

The present study suffers from some limitations, some of which were determined by its exploratory nature. First, the sample should be enlarged to allow more solid results and their replication. As already pointed out in the discussion, the low power of the study may explain some of the non-significant effects. In fact, considering the overall small β coefficients which emerged in the total effect models of the questionnaires’ main scales, a larger sample size may be required to detect an effect when there actually is one.

We also acknowledge that the choice of performing a series of simple mediation analyses may have raised the risk of an inflated Type I error. Future studies, along with enlarging the sample size, may consider the use of alternative statistical approaches, such as structural equation modeling, which may help to reduce this risk.

Second, the risk of a selection bias cannot be excluded. All BT patients were referred to our institute by the radiology unit of hospitals for further functional assessments and/or to receive rehabilitation treatments. However, the decision of accepting to take part in these activities was made by the family and thus depended on different variables, such as the geographic region of origin, the family situation (e.g., the presence of siblings) and work conditions. Indeed, this can be considered a source of selection bias. We can speculate that families in more disadvantaged conditions would find it more difficult to access our health services, in terms of time and logistical organization. Nevertheless, we were able to observe a link between SES and cognitive and psychological adjustment that may possibly become more evident with a more representative sample.

Third, all the selected psychological outcome measures were parent-reported questionnaires, which are of course subject to bias, and no self-reporting forms were examined, leading to the loss of potentially relevant information.

Moreover, PS was selected as the only cognitive mediator in the statistical analysis. Although this choice was based on clinical and literature-related preconditions, the selection of other cognitive mediators (i.e., memory) could have highlighted different aspects of the relationship between SES and psychological adjustment. The lack of additional information on the mental health status or on comorbidities also needs to be mentioned as a limitation, as it prevented us from drawing a more precise psychopathological profile of our sample.

Finally, SES is an extremely complex construct, which can be operationalized in many different ways [48,49,50,51] and which acts like a proxy of other important aspects (i.e., parent behaviors or parenting styles) that were not directly addressed in the current study. Therefore, all the inferences regarding the impact of SES made here should be approached with caution. Moreover, even though the effect of the “full” mediation of PS has been written about, it should be pointed out that this effect cannot rule out additional, relevant mediating pathways involving other factors [137] which most certainly exist, not only regarding the relation between SES and externalizing symptoms, but also regarding that existing between SES and internalizing symptoms.

## 5. Conclusions

The current study was the first, to the best of our knowledge, to investigate the relationship between SES and psychological functioning in PBTSs, considering the contribution of PS as a mediator. The results demonstrated that PS mediated the relationship between SES and the overall psychological adjustment of the patients, with a stronger mediating effect on the relationships between SES and internalizing and attention problems, whereas it did not mediate effect on the relationship between SES and conduct problems.

To conclude, PS should be carefully taken into account, especially for PBTSs who live in disadvantaged situations, when designing interventions to improve the psychological adjustment of a clinical population with such a variety of psychological sequelae, resulting not only from the diagnosis but also from the necessary oncological treatments.

## Figures and Tables

**Figure 1 cancers-14-03075-f001:**
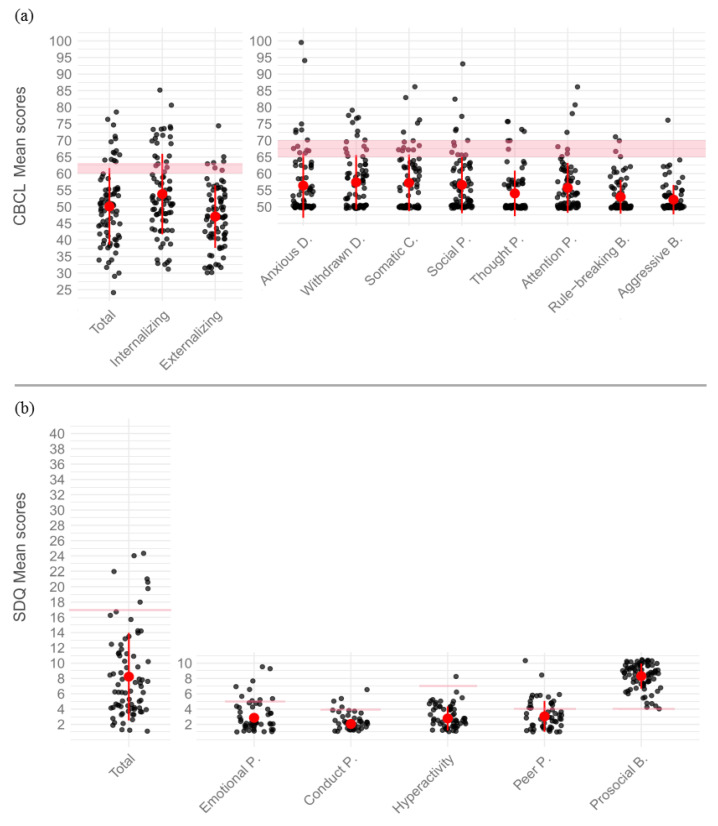
Dot plot displaying data distribution of (**a**) CBCL scores and (**b**) SDQ scores. For CBCL (**a**) the area shaded red delimits borderline values above which the score was considered to fall within the clinical range. For SDQ (**b**) the red lines indicate the threshold value, varying among scales, above which the score was considered to fall within the clinical range. For the Prosocial B. Ssale only, scores below 4 were considered in the clinical range. The red dots represent mean scores. Error bars represent ± 1 standard deviations.

**Figure 2 cancers-14-03075-f002:**
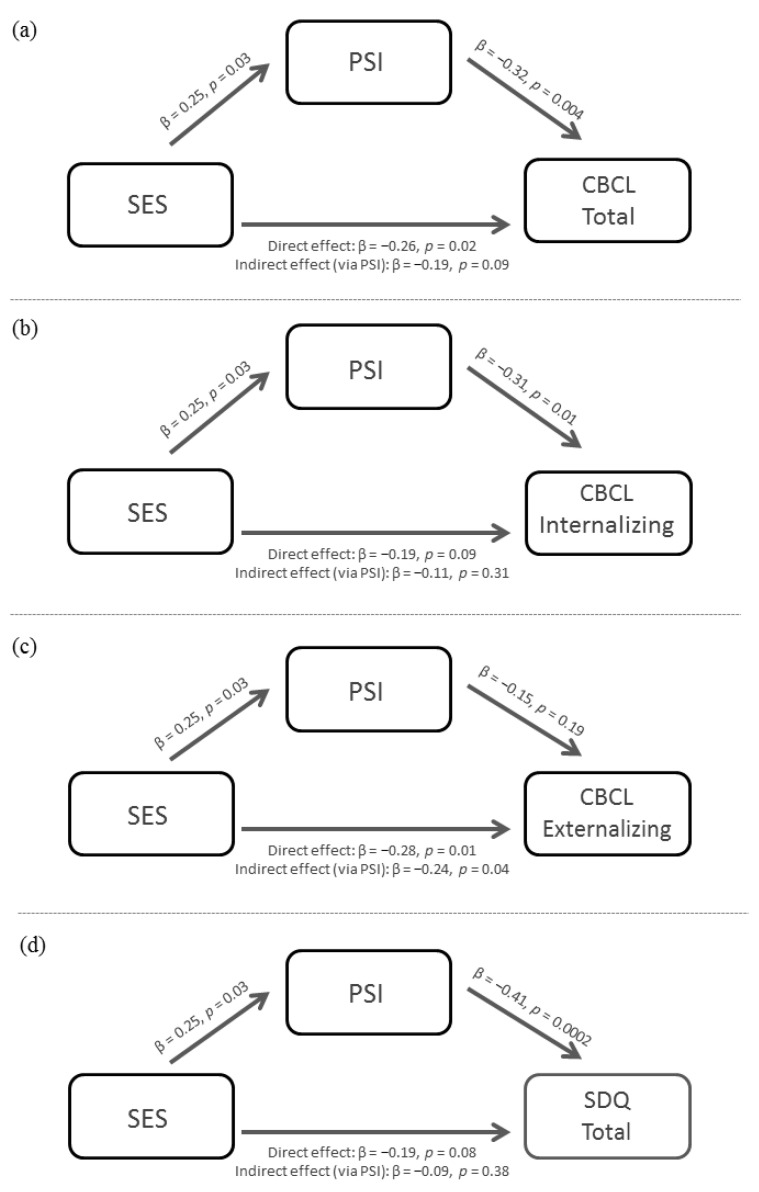
Mediation models for (**a**) CBCL total, (**b**) CBCL internalizing, (**c**) CBCL externalizing and (**d**) SDQ total scores.

**Table 1 cancers-14-03075-t001:** Number (*n*), percentage (%) or mean (M) and standard error (SE) of clinical variables of the final sample (*n* = 80) used for statistical analysis.

Categorical Clinical Variables	*n* (%)
**Sex**	
Male	45 (56.3)
Female	35 (43.7)
**Ethnicity**	
Caucasian	75 (93.75)
North-African	4 (5)
Asian	1 (1.25)
**Histopathological tumor type**	
Astrocytoma	16 (20)
Ependymoma	19 (23.7)
Medulloblastoma	20 (25)
Others	25 (31.3)
**Tumor grade**	
I	21 (26.3)
II	11 (13.7)
III	15 (18.7)
IV	29 (36.3)
NA	4 (5)
**History of hydrocephalus**	
Present	14 (17.5)
Absent	66 (82.5)
**Tumor location**	
Supratentorial	38 (47.5)
Infratentorial	42 (52.5)
**Treatment**	
Neurosurgery without adjuvant treatments	16 (20)
Neurosurgery and chemotherapy	6 (7.5)
Neurosurgery and radiotherapy +/− chemotherapy	58 (72.5)
**Continuous demographic and clinical variables**	**M (SE)**
SES	32.6 (1.2)
FSIQ	88.5 (2.3)
PSI	83.2 (2.2)
Time since diagnosis (months)	52.9 (3.7)
Age at diagnosis (months)	73.5 (4.7)

NA: grade not classifiable due to lack of biopsy information.

**Table 2 cancers-14-03075-t002:** Pearson’s r coefficient of correlations between occupation/education levels of the mother (M) and the father (F) and CBCL/SDQ main scales.

Indexes	CBCL Total	CBCL Internalizing	CBCL Externalizing	SDQ Total
Occupation level (M)	−0.21 ^	−0.13	−0.24 *	−0.26 *
Education level (M)	−0.15	−0.16	−0.18	−0.11
Occupation level (F)	−0.04	−0.02	−0.01	0.01
Education level (F)	−0.15	−0.15	−0.1	−0.13
Family occupational status	−0.21 ^	−0.13	−0.22 *	−0.22 ^
Family educational status	−0.17	−0.17	−0.16	−0.14
CBCL occupation index (M)	−0.21 ^	−0.13	−0.25 *	−0.25 *
CBCL occupation index (F)	−0.09	−0.06	−0.07	0.07

* indicates fully significant correlations (*p* < 0.05). ^ indicates marginally significant correlations (*p* < 0.09).

**Table 3 cancers-14-03075-t003:** Pearson’s *r* coefficient of correlation between SES index/PSI and CBCL/SDQ main scales and subscales.

Scales/Subscales		SES	PSI
	CBCL total	−0.26 *	−0.37 **
	Internalizing	−0.19 ^	−0.34 **
	Externalizing	−0.28 *	−0.21 ^
	SDQ total	−0.19 ^	−0.44 **
CBCL syndrome subscales	Anxious/depressed	−0.12	−0.29 *
	Withdrawn/depressed	−0.21 ^	−0.35 **
	Somatic complaints	−0.17	−0.26 *
	Social problems	−0.19	−0.40 **
	Thought problems	−0.21 ^	−0.29 *
	Attention problems	−0.28 *	−0.37 **
	Rule-breaking behavior	−0.27 *	−0.23 *
	Aggressive behavior	−0.30 *	−0.21 ^
SDQ subscales	Emotional problems	−0.07	−0.26 *
	Conduct problems	−0.14	−0.21 ^
	Hyperactivity	−0.26 *	−0.39 **
	Peer problems	−0.14	−0.44 **
	Prosocial behavior	−0.06	0.10

* indicates *p* < 0.05, ** indicates *p* < 0.01, ^ indicates marginally significant correlations (*p* < 0.09).

**Table 4 cancers-14-03075-t004:** A summary of the mediation model results for the CBCL and SDQ main scales (with covariate adjustment) and subscales (without covariate adjustment).

Scales/Subscales		Total Effect				Direct Effect				Indirect Effect			
		β	SE	CI	*p*-Value	β	SE	CI	*p*-Value	β	SE	CI	
with covariate adjustment	CBCL total	−0.25	0.13	−0.52, −0.02	0.03	−0.18	0.12	−0.42, 0.07	0.16	−0.09	0.05	−0.20, −0.006	*
	CBCL internalizing	−0.16	0.13	−0.42, 0.11	0.24	−0.06	0.13	−0.32, 0.19	0.62	−0.08	0.05	−0.21, −0.005	*
	CBCL externalizing	−0.25	0.1	−0.45, −0.05	0.02	−0.21	0.1	−0.42, −0.003	0.04	−0.05	0.04	−0.14, 0.01	
	SDQ total	−0.07	0.06	−0.19, 0.06	0.29	−0.01	0.06	−0.12, 0.11	0.92	−0.11	0.06	−0.25, −0.01	*
CBCL syndrome subscales	Anxious/depressed	−0.12	0.1	−0.31, 0.08	0.27	−0.06	0.1	−0.25, 0.15	0.62	−0.07	0.04	−0.17, −0.001	*
	Withdrawn/depressed	−0.21	0.08	−0.33, 0.01	0.06	−0.14	0.08	−0.27, 0.06	0.22	−0.08	0.05	−0.18, −0.004	*
	Somatic complaints	−0.17	0.08	−0.31, 0.04	0.14	−0.11	0.09	−0.27, 0.09	0.33	−0.06	0.05	−0.17, 0.01	
	Social problems	−0.19	0.09	−0.32, 0.03	0.09	−0.09	0.09	−0.25, 0.09	0.38	−0.09	0.06	−0.21, −0.01	*
	Thought problems	−0.21	0.07	−0.27, 0.01	0.06	−0.15	0.07	−0.23, 0.05	0.19	−0.06	0.04	−0.16, 0.003	
	Attention problems	−0.28	0.08	−0.34, −0.04	0.01	−0.19	0.08	−0.29, 0.01	0.07	−0.08	0.04	−0.18, −0.01	*
	Rule-Breaking behavior	−0.27	0.05	−0.23, −0.02	0.02	−0.22	0.05	−0.21, 0.001	0.05	−0.02	0.02	−0.06, 0.004	
	Aggressive behavior	−0.29	0.05	−0.21, −0.03	0.01	−0.26	0.05	−0.19, −0.02	0.02	−0.03	0.03	−0.11, 0.02	
SDQ subscales	Emotional problems	−0.07	0.02	−0.06, 0.03	0.51	−0.01	0.02	−0.05, 0.05	0.93	−0.06	0.05	−0.17, 0.01	
	Conduct problems	−0.14	0.02	−0.05, 0.01	0.21	−0.09	0.02	−0.04, 0.02	0.41	−0.05	0.03	−0.12, 0.004	
	Hyperactivity	−0.26	0.02	−0.08, −0.01	0.02	−0.18	0.02	−0.06, 0.01	0.11	−0.08	0.05	−0.19, −0.01	*
	Peer problems	−0.14	0.02	−0.07, 0.02	0.22	−0.03	0.02	−0.05, 0.04	0.76	−0.11	0.06	−0.24, −0.01	*
	Prosocial behavior	−0.06	0.02	−0.05, 0.03	0.62	−0.09	0.02	−0.05, 0.02	0.46	0.03	0.03	−0.02, 0.11	

Standardized coefficient (β), standard error (SE), 95% confidence interval (CI) and *p*-value of the total effect and direct effect models. For indirect effects, β coefficients, SE and 95% CI are reported; bootstrap confidence intervals crossing over zero (indicated by *) indicate significant *p*-values (i.e., *p* < 0.05).

## Data Availability

The dataset analyzed during the current study is available from the corresponding author on reasonable request.

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
