# Peer review of "The Influence of Socioeconomic Status (SES) and Processing Speed on the Psychological Adjustment and Wellbeing of Pediatric Brain Tumor Survivors"

_cancers, 2022, doi:10.3390/cancers14133075_

Round 1

Reviewer 1 Report

Overall, I believe your paper addresses an important topic, and you did a good job articulating the rationale of your study. I have a couple of suggestions for the authors to consider:

  1. In your literature review section, the conceptual association between SES, PSI, and CBCL or SDQ scores was primarily justified as an empirical relationship without an in-depth discussion on the potential theoretical association between these variables. It is important to discuss these from a theoretical standpoint.
  2. In your method section, you stated "In order to verify the assumptions to run mediation analysis, Pearson correlations were conducted for the continuous outcome variables ..." I am not exactly sure what you meant by assumptions? Distributional assumptions, relational assumptions, or others?
  3. You ran multiple mediation analyses using the four sets of data, which raises the issue of inflated Type I error. Why the authors did not use structural equation modeling (SEM) to run these multiple mediations at the same time?
  4. A discussion on power analysis should be included in your method section. 
  5. Am I reading it right that the only covariate you included in your models was the age at diagnosis, and the justification here was that age was the only significant predictor? What about sex, race, and clinical diagnosis, all of which are important demographic and clinical covariates to be included. 
  6. Many of your discussions on non-significant findings were for theoretical reasons, but I think the low power of your study probably is a main driver.

Author Response

Please, see reply to Reviewer 1 in the attach.

Reviewer 2 Report

The authors have presented data from a single-institution, retrospective sample of brain tumor survivors examining the relationship between socioeconomic status, processing speed and psychological functioning. While the results are interesting, I have a number of concerns.

Methods and Statistical Analysis: 

1) Please clarify what proportion of brain tumor survivors are admitted to the pediatric rehabilitation center? Is this affiliated with a pediatric cancer center. I am concerned that it may not be representative of pediatric brain tumor survivors overall.

2) It is unclear to me the number of patients who were potentially eligible to be included and the reasons for their exclusion. Why were only ~2/3 of the patients from the 2 prior studies cited in lines 151 and 152 included? How many potential patients were missing the measures used? 

3) Why was only processing speed evaluated for mediation and not, for example, memory, which is often impacted and assessed during neuropsychological testing? 

4) Overall, the methods section is very long and while organized by section there are areas that seem more well suited for the results or discussion section. Specifically areas like P5 under 2.2.3 and in-depth discussion of the modification of the SES scale 2.2.5. 

The presentation of the correlation results in the methods section seemed out of place (Table 2 and Table 3).

5) Was there any data on treatment of psychological or mental health conditions in patients (such as depression, anxiety or ADHD) and was this considered in the analysis?

Results:

6) Please clarify included covariates for presented results of mediation analysis. I did not see mention of sex as a covariate and am curious if it was used. 

Discussion: 

7) Any insight into the potential efficacy of PS intervention in the setting of lower SES families or at what time point they should occur (by age, by time from diagnosis etc.) may strengthen the discussion.

8) Discussion of the clinical significance of the statistically significant findings would be useful, what effect size might matter for an individual patient? 

Author Response

Please, see reply to Reviewer 2 in the attach.

Round 2

Reviewer 1 Report

While there is nothing wrong with simple mediation models, I think if the authors insisted on running multiple mediations instead of SEM, you should acknowledge this as a limitation of your study.

Author Response

While there is nothing wrong with simple mediation models, I think if the authors insisted on running multiple mediations instead of SEM, you should acknowledge this as a limitation of your study.

Response : Following the reviewer’s indication, we added this point in the limitations section, as follows:

“We also acknowledge that the choice of performing a series of simple mediation analyses may have raised the risk of inflated Type I error. Future studies, along with enlarging the sample size, may consider the use of alternative statistical approaches, such as structural equation modeling, that may help reduce this risk.”

Reviewer 2 Report

The authors have appropriately responded to prior reviewer suggestions which have improved the manuscript. They have also addressed additional limitations including the possibility of selection bias and that although, the use of processing speed is understandable, there was not assessment of other cognitive measures.

Author Response

The authors have appropriately responded to prior reviewer suggestions which have improved the manuscript. They have also addressed additional limitations including the possibility of selection bias and that although, the use of processing speed is understandable, there was not assessment of other cognitive measures.

Response : We thank the reviewer for his/her comments, which have helped us to improve the manuscript.